# The ALCHEmist: Automated Labeling 500x CHEaper Than LLM Data Annotators

**Tzu-Heng Huang, Catherine Cao, Vaishnavi Bhargava, Frederic Sala**
University of Wisconsin-Madison
{thuang273, ccao35, vbhargava3}@wisc.edu,
fredsala@cs.wisc.edu

## Abstract

Large pretrained models can be used as annotators, helping replace or augment crowdworkers and enabling distilling generalist models into smaller specialist models. Unfortunately, this comes at a cost: employing top-of-the-line models often requires paying thousands of dollars for API calls, while the resulting datasets are static and challenging to audit. To address these challenges, we propose a simple alternative: rather than directly querying labels from pretrained models, we task models to *generate programs that can produce labels*. These programs can be stored and applied locally, re-used and extended, and cost orders of magnitude less. Our system, **Alchemist**, obtains comparable to or better performance than large language model-based annotation in a range of tasks for a fraction of the cost: on average, improvements amount to a **12.9%** enhancement while the total labeling costs across all datasets are reduced by a factor of approximately **500×**. We release our code here: `https://github.com/SprocketLab/Alchemist`.

## 1 Introduction

One of the most exciting developments in machine learning is the use of large pretrained models to act as *annotators* or *labelers* [1, 2, 3, 4, 5, 6, 7, 8]. This includes the use of large language models (LLMs) like GPT-4 [9] and Claude 3 [10]. This process offers multiple benefits. First, pretrained models are an efficient way to annotate and have the potential to partially or fully replace expensive human crowdworkers [2, 6, 11, 12]. Second, this approach allows for *distilling* large models into smaller, task-specific models that can be deployed locally at lower cost [3, 13, 7, 8]. This is additionally important in settings like healthcare and finance where privacy laws require the use of local models.

Despite this promise, pretrained model-based annotation has several drawbacks that stymie its adoption. These drawbacks include

- **High Cost**: Labeling a dataset can be expensive. This is particularly so in cases where each data point consists of many tokens. For example, we find that labeling a moderately-sized dataset [14] with 7,569 data points using GPT-4 costs over $1,200.

- **Lack of Extensibility**: Making even small changes to specifications necessitates re-running the entire pipeline to obtain new labels. This inflexibility means the resulting labels are static.

- **Inability to Audit**: API access to pretrained models does not permit inspecting most aspects of the model. Users must simply accept the provided labels with only minimal additional information. Techniques that ask the model for explanations for its decisions may not be reliable [15, 16, 17].

We address these obstacles through a simple but surprisingly powerful notion. Rather than having pretrained models label data, we task language models to ***generate programs that can output labels***. These synthesized programs serve as annotators, capturing the underlying logic used by the models when annotating. In other words, instead of distilling a powerful model to label a dataset (and

38th Conference on Neural Information Processing Systems (NeurIPS 2024).

**Prompt Template in Alchemist**

| | |
|---|---|
| *[Task Description]* | ‣ Write a bug-free and executable function in python to label comment on Youtube as spam or ham. |
| *[Labeling Instruction]* | ‣ If the comment is spam, function returns 1. If the comment is ham, function returns 0. |
| | ‣ If the comment cannot be determined as spam or ham, function returns -1. |
| *[Function Signature]* | ‣ def label_spam (text_comment): |
| *[Task Description]* | ‣ Write a bug-free and executable function in python to label the topic of biomedical document. |
| *[Labeling Instruction]* | ‣ If document is about colon cancer, returns 0. If document is about lung cancer, returns 1. |
| | ‣ If document is about thyroid cancer, returns 2. If document cannot be categorized, returns -1. |
| *[Function Signature]* | ‣ def label_cancer (document): |

```python
def label_spam(text_comment):
    # Define spammy characteristics
    url_pattern = r'http[s]?://(?:[a-zA-Z]|[0-9]|[$-_@.&+]|[!*\(\),]|(?:%[0-9a-fA-F][0-9a-fA-F]))+'
    email_pattern = r'\S+@\S+'
    phone_pattern = r'\(?(?[0-9]{3})\)?([ .-]?)([0-9]{3})\2([0-9]{4})'
    multiple_exclamations = r'!!+'
    multiple_questions = r'\?\?+'
    keyword_list = ['subscribe', 'channel', 'check out', 'follow', 'my page', 'discount', 'save', 'free', 'buy', 'purchase']

    # Check if comment has URL, email, or phone number
    if re.search(url_pattern, comment) or re.search(email_pattern, comment) or re.search(phone_pattern, comment):
        return 1

    # Check if comment has multiple exclamation marks or question marks
    if re.search(multiple_exclamations, comment) or re.search(multiple_questions, comment):
        return 1

    # Check if comment has spammy keywords
    for keyword in keyword_list:
        if keyword in comment.lower():
            return 1

    # If none of these conditions are met, return 0 (ham)
    return 0
```

```python
def label_cancer(document):
    # define key terms for each cancer type
    colon_cancer_terms = ['polyp', 'colonoscopy', 'rectum', 'colorectal', 'colectomy']
    lung_cancer_terms = ['bronchoscopy', 'smoking', 'lung', 'pulmonary', 'lobectomy']
    thyroid_cancer_terms = ['thyroid', 'thyroxine', 'goiter', 'thyroidectomy']

    # convert document to lowercase
    document = document.lower()

    # check for key terms in document
    colon_cancer = any(term in document for term in colon_cancer_terms)
    lung_cancer = any(term in document for term in lung_cancer_terms)
    thyroid_cancer = any(term in document for term in thyroid_cancer_terms)

    # classify document based on presence of key terms
    if colon_cancer and not (lung_cancer or thyroid_cancer):
        return 0
    elif lung_cancer and not (colon_cancer or thyroid_cancer):
        return 1
    elif thyroid_cancer and not (colon_cancer or lung_cancer):
        return 2
    else:
        return -1  # document cannot be categorized or belongs to multiple categories
```

Figure 1: Examples of generated programs and their prompts. These are synthesized by GPT-4 for spam detection and cancer identification tasks. Programs use regular expressions (left program) and keyword matching (right program) as their labeling logic to classify data points.

subsequently training a smaller model on the labeled data), we *distill directly into code* (Figure 1). These resulting programs can either make predictions directly or can label training dataset then train a downstream model using it[1].

This simple notion resolves all of the challenges related to pretrained model-based annotation. First, *API calls scale with the number of programs instead of the number of data points.* That is, since we generate programs that can themselves make any number of predictions locally at no cost, we can reduce the number of API calls by orders of magnitude. For example, for the dataset described above [14], the number of GPT-4 calls was reduced from 7,569 (the size of the dataset) to 10 (the number of generated programs), resulting in a massive cost reduction from \$1,200 to \$0.70, a 1,700-fold decrease. Moreover, code can be easily inspected, corrected, and extended, allowing seamless adaptation when prediction classes or labeling rules change.

While a powerful idea, distilling model into code presents several challenges. First, any particular program may be inaccurate, fail to compile, or may otherwise be flawed, resulting in noisy program outputs. We address this obstacle by applying *weak supervision*, a framework for dataset construction from multiple noisy sources of signal [18, 19, 20, 21]. Next, operating on non-text modalities is challenging. We handle this via a simple two-step approach that first extracts high-level concepts and then uses them in concert with a local feature extractor to enable tractable program generation.

**Contributions.** We propose an alternative approach to replace expensive annotation processes that require repetitive prompting for labels. We developed a system called *Alchemist* that implements this idea. Empirically, Alchemist improves performance five out of eight datasets, with an average enhancement of 12.9%—while reducing total costs by a factor of approximately $500\times$. Finally, we introduce and validate extensions that address non-text modalities.

## 2 Related Work

Our work relates to LLM-based annotation, prompting, and the weak supervision framework.

**Using Large Pretrained Models for Data Annotation.** Large pretrained models have demonstrated powerful capabilities using zero-shot prompting across a wide range of tasks [1]. One promising development is their potential to serve as data labelers, which can reduce the cost and human effort in

---

[1]The latter option is preferable, as these models can often generalize beyond their source of supervision [13]

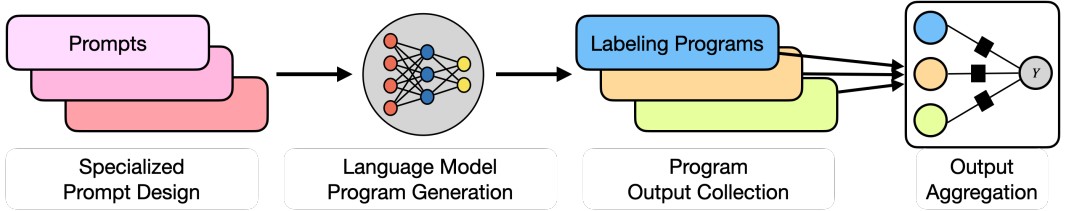

Figure 2: Overall workflow for Alchemist.

data labeling [1, 2, 11, 12]. Existing research in this area mainly focuses on approaches that allow for more efficient inference, enhanced label generation, and distilling into smaller but specialized labelers [3, 4, 5, 6, 7, 8]. However, *scalability* is the main limitation in these approaches, as making inferences via querying an API for data examples can be cost-prohibitive. To tackle this challenge, rather than prompting for labels repetitively, we propose prompting pretrained models for programs that use synthesized labeling logic and can thus serve as alternative data labelers.

**Prompt Engineering & In-Context Learning.** In-context learning adapts pretrained models to new tasks without additional fine-tuning [1]. It involves providing relevant examples as demonstrations to solve the task, such as pairs of languages for translation [22]. By including task-specific examples, models can better understand the task at hand. Adding a few data points as demonstrations [23] is commonly suggested when models act as data annotators. Moreover, they can be selected [24, 25], retrieved [26], or more efficiently, generated [27]. We explore various types of supplementary information that can be added to Alchemist to help improve program generation and permit more control over the labeling logic used in the programs.

**Weak Supervision Framework.** Weak supervision enables the rapid creation of large training datasets by aggregating cheap-but-noisy signals derived from various labeling sources [18, 19, 21, 28]. These sources can be crafted by domain expertise, using labeling heuristics, or even trained on smaller, weaker classifiers [29, 30, 31, 32]. Recent advancements in code generation open up the potential to automate the heuristic design process. Frameworks such as ScriptoriumWS [33], and DataSculpt [34] have been developed to take advantage of code-generating models [35, 9, 36] to craft weak supervision sources through prompting. While similar in spirit to our approach, these have several drawbacks: ScriptoriumWS requires more human effort in prompt engineering to better guide code-generation models. Both ScriptoriumWS and DataSculpt can perform poorly in tasks requiring specific domain expertise and, most importantly, they do not handle modalities beyond text—unlike Alchemist.

## 3 Alchemist System

We begin by presenting a general annotation workflow in Alchemist, followed by a detailed discussion of each key step.

**General Workflow.** The process is depicted in Fig. 2. First, users select an unlabeled dataset and create simple prompts to instruct language models to generate programs that incorporate labeling logic. These prompts can integrate relevant information and may vary in their design, allowing for the synthesis of multiple programs. Next, given a set of generated programs and their outputs, we apply weak supervision techniques to obtain a set of aggregated labels. Finally, the labeled points can be used to train a distilled model that can be stored and used locally.

### 3.1 Prompting Strategy

We propose a general and extensible prompt template for querying language models to generate annotator programs. This general template consists of three key components:

- **Task Description**: Provides the model an overview of generated program's desired objectives.
- **Labeling Instructions**: Specifies classes and the expected structure of the program's output.
- **Function Signature**: Describes the function's name and the input types to be used.

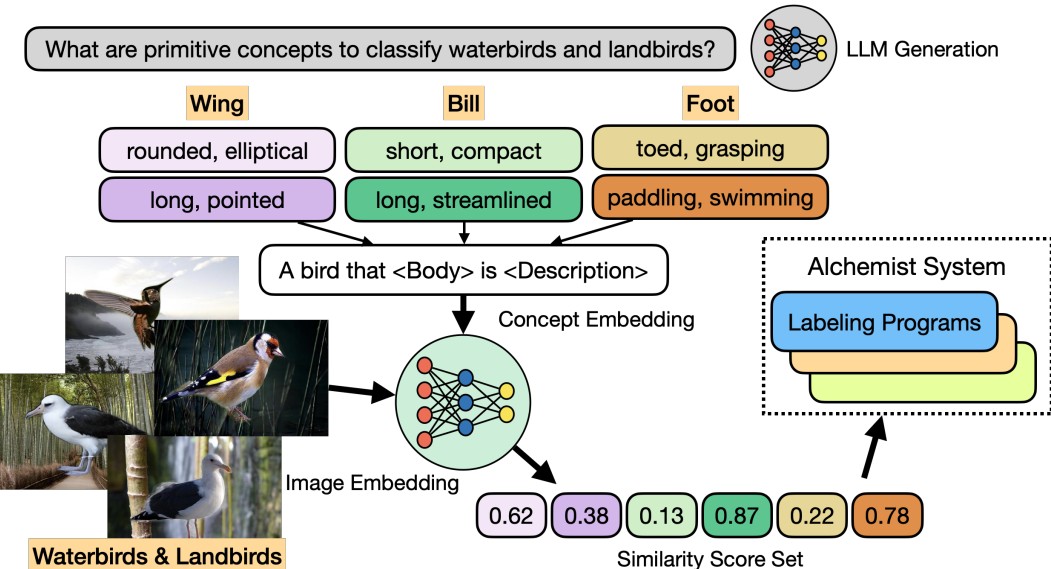

Figure 3: Alchemist can handle rich modalities through a simple extension. First, a language model identifies task-specific concepts (top). Then, a local multimodal model is used as a feature extractor for these concepts, producing low-dimensional feature vectors that can be ingested by generated labeling programs.

This simple but general template allows for flexible incorporation of various types of information, enabling the model to generate programs that are tailored to specific requirements. Two sample prompt templates in Alchemist are displayed in Fig 1.

**Using Supplementary Information.** Drawing inspiration from few-shot prompting [37, 1], where users provide demonstrations (i.e., data points with their labels) to enhance generated responses, we explore various types of supplementary information that can be integrated to assist models in synthesizing programs. This approach is particularly useful for scenarios where language models may lack the expertise to generate effective programs, or where specific adaptations in labeling logic are required. Such information can be crafted by users themselves, domain experts or, more efficiently, generated by language models themselves. Additionally, it can be combined with retrieval-augmented generation (RAG) systems [38, 39] to access external knowledge.

We explore various types of supplementary information to assist in code generation, starting with high-level concepts and then progressively looking into more practical details to control programs.

*Dataset and Prediction Class Description.* First, supplementary information can include relevant background details about the purpose for which the dataset was built and high-level information about the dataset, such as definitions for each label class. By providing this context, the language model can better understand the task at hand.

*Data Exemplars.* Furthermore, we recommend including a small number of labeled data examples in the prompt. This can help language models better comprehend the specific problem. Examples act as concrete illustrations of the task, offering a clearer understanding of the expected output. This can be particularly beneficial when dealing with a complex problem.

*Keywords.* Next, labeling logic in programs can make use of keyword-searching techniques (e.g., Fig 1). For instance, in situations such as spam detection or topic classification, certain words or phrases may have a strong correlation with specific classifications. Providing several keywords in the prompt may lead models to create labeling programs that explicitly search for the presence or absence of these keywords. This allows for more targeted and precise labeling.

*Specialized Labeling Rules.* Finally, more prior knowledge such as heuristics, specialized labeling rules, guidance, and domain-specific knowledge can be integrated into the prompt. This information can provide concrete labeling steps on how to label specific classes and offer greater control over the logic implemented in the generated programs.

```
def label_bird_image(image):
    """
    Heuristic function to classify images of birds as landbirds or waterbirds.
    Returns:
    - str: "landbird" or "waterbird" based on heuristic analysis.
    """
    img = np.array(image)

    # Assuming more green indicates land surroundings.
    # Assuming more blue indicates water surroundings.
    green_threshold, blue_threshold = 50, 100

    green_pixels = np.sum((img[:,:,1] > green_threshold) & \
                          (img[:,:,0] < green_threshold) & \
                          (img[:,:,2] < green_threshold))
    blue_pixels = np.sum(img[:,:,2] > blue_threshold)

    if blue_pixels > green_pixels:
        return "waterbird"
    else:
        return "landbird"
```

```
def label_bird_image(toed_grasping_score, paddling_swimming_score):
    """
    Labels bird images into classes based on foot type similarity scores.
    Parameters:
    - toed_grasping_score (float): Similarity score for 'toed, grasping'.
    - paddling_swimming_score (float): Similarity score for 'paddling, swimming'.
    Returns:
    - str: "landbird", "waterbird", or -1 if it cannot be determined.
    """
    threshold = 0.5
    if toed_grasping_score > threshold and paddling_swimming_score < threshold:
        return "landbird"
    elif paddling_swimming_score > threshold and toed_grasping_score < threshold:
        return "waterbird"
    elif abs(toed_grasping_score - paddling_swimming_score) < 0.1:  # Similar scores
        return -1
    else:
        if toed_grasping_score > paddling_swimming_score:
            return "landbird"
        else:
            return "waterbird"
```

Figure 4: Program examples generated by GPT4o on Waterbirds dataset. The left program is synthesized by directly asking for a labeling program when the input is an image (raw pixels), while the right program uses Alchemist's extension. The former labels birds using the dominant color in the image, which can be predicted incorrectly due to spurious correlations (e.g., background).

Overall, supplementary context is provided before the task description to enhance language models' understanding of the task. This, in turn, enables models to generate programs that are more effective and tailored to the specific requirements of user needs.

## 3.2 Dataset Synthesis

While generated programs can efficiently annotate data, these programs may produce outputs that are noisy or inaccurate. However, as such programs may employ different techniques, such as pattern-matching, heuristic rules, or other approaches—each with its own strengths and limitations—there may be *complementary* signal in their outputs. This means we can aggregate them to mitigate the impact of noise. To do so, we apply weak supervision techniques [18, 19, 20, 21]. This process starts by learning a model of the reliabilities of the programs. Once learned, this model enables aggregating label outputs from different programs into high-quality *pseudolabels*.

Alchemist is compatible with a variety of weak supervision aggregation models, called *label models*, providing flexibility in the choice of the weak supervision approach. For simplicity, in this work, we focus on using the Snorkel framework [19], which is a standard and widely-used approach in the weak supervision community.

## 3.3 Extensions: Handling Complex Modalities.

Crafting programs that operate over text is relatively easy for large language models. More complex data modalities, however, can be far more challenging. Consider images as an illustrative example. Even employing state-of-the-art multimodal models, e.g., GPT-4o [40] and GPT-4V [9], to seek programs operating over sample images may not produce satisfactory results.

To address this challenge, we extend Alchemist's pipeline to include an intermediate step. Specifically, we convert the raw data (i.e., in our example, image pixels) into a set of features representing high-level concepts. These concepts are obtained by prompting a language model (or, potentially, a multimodal model) to identify task-relevant notions. For example, for a bird categorization task, models may identify "wing shape," "beak shape," or "foot type" as informative concepts for distinguishing between bird species. Next, we use any open-source local multimodal model, like CLIP [41], as a feature extractor for the identified concepts, producing low-dimensional feature vectors that can be easily ingested by generated programs. As such models are free, this does not increase our cost.

Fig. 3 and Fig. 4 present examples of generated high-level concepts and the corresponding programs used for the Waterbirds dataset, where the task is to distinguish between landbird and waterbird specices [42]. This simple approach can be applied to any data modality where we have access to a local multimodal model (i.e., a model operating on the modality of interest and text).

|  | YouTube | | SMS | | Yelp | | IMDb | |
| --- | --- | --- | --- | --- | --- | --- | --- | --- |
|  | Est. Cost | Accuracy | Est. Cost | F1-score | Est. Cost | Accuracy | Est. Cost | Accuracy |
| Zero-shot Prompting | 0.096 | 0.871 | 0.240 | **0.907** | 3.873 | **0.845** | 3.400 | **0.737** |
| Alchemist with GPT-3.5 | 0.004 | **0.891** | 0.004 | 0.900 | 0.005 | 0.575 | 0.004 | 0.662 |
|  | MedAbs | | Cancer | | Finance | | French | |
|  | Est. Cost | Accuracy | Est. Cost | Accuracy | Est. Cost | Accuracy | Est. Cost | Accuracy |
| Zero-shot Prompting | 1.944 | 0.311 | 15.925 | 0.716 | 0.201 | 0.641 | 0.641 | 0.611 |
| Alchemist with GPT-3.5 | 0.006 | **0.346** | 0.003 | **0.968** | 0.007 | **0.660** | 0.006 | **0.690** |

Table 1: Testing performance of the distilled model is reported for each combination of method and dataset. The estimated cost is obtained by calculating the number of input and output tokens associated with GPT-3.5's pricing table [49]. Other models may be even more expensive.

# 4 Experiments

We study the capability of Alchemist empirically. Our goals are to validate the following claims:

- **Cost Reduction and Improved Performance (Sec. 4.1):** Alchemist can reduce cost by orders of magnitude, while producing labels of similar or better accuracy.

- **Extendibility to Other Modalities (Sec. 4.2):** Alchemist can operate with modalities beyond text.

- **Use of Supplementary Information (Sec. 4.3):** Incorporating relevant information into prompts enables the generation of better programs, yielding more accurate pseudolabels.

- **More Diverse Programs Can Help (Sec. 4.4):** Increasing the diversity of generated programs created by different labeling logic enables better pseudo labels.

- **Comparing to Human-crafted Programs (Sec 4.5):** Synthesized programs may be more effective in comparison to human-crafted ones.

**Datasets.** We include diverse datasets covering text and image modalities. For text, we include eight datasets that span three different types of language tasks. These include the YouTube [43], SMS [44] datasets for spam classification, IMDb [45], Yelp [45], Finance [46], and French [47] datasets for sentiment analysis, and the MedAbs [48] and Cancer [14] datasets for topic classification. We note that the Finance, French, MedAbs, and Cancer datasets are relatively challenging, with points that require a degree of domain expertise for accurate labeling. For example, the French dataset requires a good understanding of the language. These may pose challenges for pretrained models.

For our extensions to richer modalities, we focus on image tasks. Our evaluation uses the Waterbirds dataset [42]. This dataset is designed to assess models' robustness to spurious correlations and ability to handle distribution shifts. More details are in Appendix A.

## 4.1 Cost Reduction and Improved Performance

**Setup.** We open our evaluation of Alchemist with text domain datasets and use GPT-3.5 to generate programs. For each dataset, we input pure prompts without supplementary information into GPT-3.5 and generate 10 programs to use. We construct training datasets by aggregating the programs' outputs into pseudolabels with the weak supervision framework Snorkel [19]. We then train a two-layer MLP as a distilled model. We run five times with different random seeds and report their average performance. As our main baseline, we directly use language models to produce annotations per point. The resulting labels are used to train a distilled model for comparison. The prompt template used in our baseline approach and our training settings are provided in Appendix A.

**Expected Results.** We anticipate that Alchemist can generate programs that can produce accurate labels while substantially reducing the expense of API calls.

**Results.** Table 1 presents the distilled model's performance on each testing dataset. We observe that label accuracy is improved on five out of eight datasets, particularly in challenging settings such as the MedAbs, Cancer, and French datasets, outperforming the baseline zero-shot prompting approach. We also report the estimated costs of building training datasets. The costs for zero-shot prompting depend on the number of tokens for the dataset. In contrast, Alchemist only prompts 10 programs for

| Feature Extractor | Method | Average Accuracy (↑) | Worst Group Accuracy (↑) | Gap (↓) |
|---|---|---|---|---|
| — | **Vanilla Alchemist with GPT4o** | 0.395 | 0.367 | 0.028 |
| | **Vanilla Alchemist with Claude 3** | 0.781 | 0.022 | 0.759 |
| CLIP ViT-B/32 | Zero-shot Prompting | 0.820 | 0.318 | 0.502 |
| | Group Prompting | **0.823** | 0.383 | 0.440 |
| | **Alchemist with GPT4o** | 0.805 | 0.283 | 0.522 |
| | **Alchemist with Claude 3** | 0.774 | **0.463** | **0.410** |
| CLIP ViT-L/14 | Zero-shot Prompting | **0.904** | 0.335 | 0.569 |
| | Group Prompting | 0.791 | 0.240 | 0.551 |
| | **Alchemist with GPT4o** | 0.802 | **0.467** | **0.335** |
| | **Alchemist with Claude 3** | 0.737 | 0.346 | 0.391 |

Table 2: Alchemist on non-text modalities. We experiment with standard Alchemist (top), our proposed extension with two CLIP-based local models as feature extractors, and CLIP prompting baselines. Alchemist achieves comparable performance on average accuracy while improving robustness to spurious correlations.

each task, resulting in a significant reduction in the costs—by orders of magnitude. This efficiency is the main advantage of Alchemist, ***as it allows for the creation of high-quality datasets with minimal expense.*** We include ablation studies with other weak supervision models within the Alchemist framework in Appendix C. They successfully demonstrate the flexibility and robustness of using Alchemist.

## 4.2 Extending Alchemist to Other Modalities

**Setup.** Next, we validate the extension of Alchemist to richer modalities. We consider our approach, where we prompt a multimodal model such as GPT4o and Claude 3, to generate high-level task-specific concepts. We extract features for these concepts by employing CLIP as our local feature extractor. This converts raw pixels into feature vectors for the extracted high-level concepts, producing a set of similarity scores. Armed with these scores, we describe scores associated with their concepts in prompts and ask GPT4o and Claude 3 for 10 programs. As before, we use Snorkel as our aggregation procedure.

*Baselines.* We study two baselines. The first is the vanilla version of Alchemist, where we directly ask GPT4o and Claude 3 to produce code that can operate on images (see left program in Fig. 4). The second is simple zero-shot prompting using CLIP, along with a variant, a group prompting approach that assumes access to spurious information and adds it to the given prompt[2].

**Expected Results.** We expect employing our two-step process can enable tractable program generation. In addition, we hypothesize that programs generated in this way are beneficial in targeting salient concepts and reducing the impact of irrelevant or shortcut features, thereby enhancing robustness.

**Results.** We present results in Table 2. Our evaluation focuses on three key metrics: average accuracy, worst group accuracy, and the gap between these two measures. Ideally, a robust model should achieve high average accuracy and high worst group accuracy while minimizing the disparity between the two. First, we see that directly asking programs to use may have very low performance (GPT4o) or may hugely suffer from spurious correlations, destroying worst group performance (Claude 3, CLIP zero-shot). Our method addresses both cases. Compared to baseline methods, Alchemist demonstrates increased worst group accuracy and a reduced gap between the average and worst group accuracies. This is a key strength of Alchemist: ***targeting salient concepts to be used as features may help move models away from spurious shortcuts found in the data***. This validates Alchemist's ability to handle complex modalities while improving robustness.

| | YouTube | | | SMS | | | Yelp | | | IMDb | | |
|---|---|---|---|---|---|---|---|---|---|---|---|---|
| | GPT-3.5 | GPT-4 | Claude 3 | GPT-3.5 | GPT-4 | Claude 3 | GPT-3.5 | GPT-4 | Claude 3 | GPT-3.5 | GPT-4 | Claude 3 |
| General Prompt | 0.92 | 0.92 | 0.66 | 0.64 | 0.62 | 0.75 | 0.65 | 0.82 | 0.78 | 0.71 | 0.77 | 0.77 |
| + Dataset Description | 0.64 | **0.93** | **0.71** | 0.63 | **0.63** | **0.76** | **0.72** | 0.82 | **0.79** | 0.70 | **0.79** | 0.73 |
| + 5 Data Exemplars | 0.91 | 0.86 | **0.76** | 0.46 | **0.66** | 0.62 | **0.72** | 0.82 | **0.82** | 0.68 | 0.75 | 0.73 |
| + Keywords | 0.76 | **0.93** | 0.53 | 0.40 | 0.42 | 0.64 | **0.69** | 0.81 | 0.78 | 0.69 | **0.78** | 0.72 |
| + Labeling Rules | 0.74 | 0.82 | 0.56 | **0.67** | **0.67** | 0.58 | **0.75** | 0.81 | **0.79** | 0.71 | 0.77 | 0.74 |
| | MedAbs | | | Cancer | | | Finance | | | French | | |
| | GPT-3.5 | GPT-4 | Claude 3 | GPT-3.5 | GPT-4 | Claude 3 | GPT-3.5 | GPT-4 | Claude 3 | GPT-3.5 | GPT-4 | Claude 3 |
| General Prompt | 0.52 | 0.53 | 0.55 | 0.71 | 0.73 | 0.59 | 0.66 | 0.49 | 0.56 | 0.65 | 0.55 | 0.56 |
| + Dataset Description | 0.49 | 0.50 | 0.51 | 0.59 | 0.62 | **0.60** | 0.61 | **0.63** | 0.62 | 0.39 | **0.58** | **0.67** |
| + 5 Data Examples | **0.53** | **0.54** | 0.55 | 0.55 | 0.57 | **0.63** | 0.60 | 0.50 | **0.60** | 0.40 | **0.69** | 0.44 |
| + Keywords | **0.55** | **0.55** | 0.55 | 0.55 | 0.55 | 0.46 | 0.66 | **0.62** | **0.65** | **0.69** | 0.66 | **0.67** |
| + Labeling Rules | 0.52 | **0.55** | **0.56** | 0.61 | 0.59 | **0.63** | 0.66 | **0.56** | **0.67** | 0.65 | **0.66** | 0.33 |

Table 3: Testing performance of the label model is reported for each combination of prompting strategy and dataset. We observe that GPT-4 and Claude 3 (that may possess better comprehension capabilities) exhibit greater enhancements when provided with supplementary information.

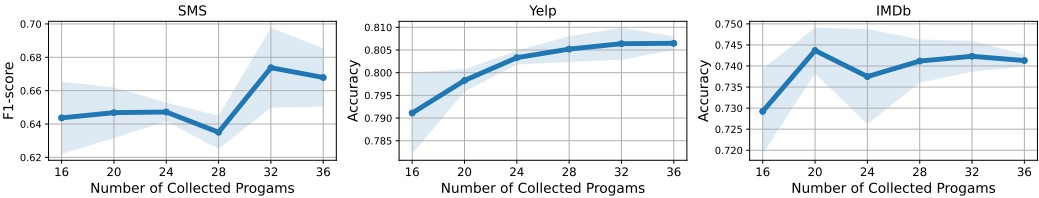

Figure 5: Performance is reported using their average performance and standard deviations. Results indicate that the label model is improved when the number of diverse programs increases.

### 4.3 Use of Supplementary Information

**Setup.** We test how integrating relevant information into the prompt context can augment generated programs. Instead of manually crafting supplementary information, we harness the power of language models to generate and integrate. This approach is useful for challenging datasets where users may not have the necessary knowledge or expertise to start. We evaluate the effectiveness of this approach, by comparing label model performance using programs generated by two different methods: pure prompting and in-context prompting. In-context prompting involves supplementary information, while pure prompting relies solely on the task description without any additional guidance. We employ GPT-3.5, GPT-4 and Claude 3 as our program sources and synthesize ten for each strategy.

**Expected Results.** We hypothesize that providing supplementary information can enhance task understanding, demonstrate specific labeling logic, and offer concrete steps, ultimately leading to better programs for use.

**Results.** Table 3 presents this comparative analysis on label model performance using different type of information. We observe that by incorporating supplementary information into pure prompts, Alchemist can guide language models to generate more effective programs, which in turn produce more accurate pseudolabels. Improvements are particularly evident in the challenging datasets such as Finance and French. Moreover, this approach can be combined with RAG systems to include external knowledge bases and customize the relevant information. Such flexibility compared to zero-shot prompting is another key strength of Alchemist, as ***programs can easily be adapted, augmented, and specialized.***

### 4.4 More Diverse Programs Can Help

**Setup.** As shown in Table 3, incorporating different supplementary information results in varying degrees of additional improvement. Potentially, certain sets of supplementary information allow the

---

[2]the group prompts are "waterbird on water background", "waterbird on land background", "landbird on water background", and "landbird on land background".

| | YouTube | | | | SMS | | | | Yelp | | | | IMDb | | | |
|---|---|---|---|---|---|---|---|---|---|---|---|---|---|---|---|---|
| | Human crafted | Synthesized Programs | | | Human crafted | Synthesized Programs | | | Human crafted | Synthesized Programs | | | Human crafted | Synthesized Programs | | |
| | | GPT-3.5 | GPT-4 | Claude 3 | | GPT-3.5 | GPT-4 | Claude 3 | | GPT-3.5 | GPT-4 | Claude 3 | | GPT-3.5 | GPT-4 | Claude 3 |
| Num. of Programs | 10 | 10 | 10 | 10 | 73 | 10 | 10 | 10 | 8 | 10 | 10 | 10 | 5 | 10 | 10 | 10 |
| Coverage | 0.89 | 1.00 | 1.00 | 1.00 | 0.41 | 1.00 | 1.00 | 1.00 | 0.83 | 0.78 | 0.99 | 0.88 | 0.88 | 0.89 | 1.00 | 0.98 |
| Performance | 0.85 | **0.89** | **0.89** | 0.72 | 0.89 | **0.90** | **0.93** | 0.89 | 0.76 | 0.57 | **0.82** | **0.83** | 0.73 | 0.66 | **0.75** | 0.70 |

Table 4: Analysis showing that Alchemist can achieve comparable or better accuracy and higher coverage while using fewer programs to label the data.

model to specialize better on certain data points than others. We seek to achieve these performance improvements *without* the need to re-prompt the model with each set of supplementary information. Instead, we collect previously generated programs to obtain a set of programs with greater diversity. We ask: *can Alchemist achieve better performance by modeling more diverse programs?*

We randomly select a set of programs from each category, collect them, and train the label model with their program outputs. Additionally, we increase the number of sampled programs in each category from 4 to 9. We test this approach on the datasets where Alchemist gives comparable or lower performance than zero-shot prompting in our initial experiments in Table 1, namely the SMS, Yelp, and IMDb datasets.

**Expected Results.** By obtaining more diverse programs to use, Alchemist can capture a wider range of perspectives and labeling logic, potentially leading to more accurate pseudolabels.

**Results.** Fig. 5 visualizes the effect on the label model's performance when we increase the diversity in collected programs. It demonstrates a trend and indicates that involving a more diverse set of programs can help to mitigate the impact of individual strategy biases or limitations, leading to the production of better labels.

Overall, results in Sec. 4.3 and in Sec. 4.4 underscore that *the use of supplementary information and involving diverse types of programs can help achieve better performance.*

## 4.5 Comparing to Human-crafted Programs

**Setup.** Lastly, we compare synthesized programs in Alchemist and manually crafted labeling functions in WRENCH [50], which is a widely-used benchmark for evaluating weak supervision methods. We focus on the datasets that overlap between Alchemist and WRENCH. For each dataset, we use pure prompts to query GPT-3.5, GPT-4, and Claude 3 for 10 programs. We then evaluate the performance of the distilled model for both methods. We also include the label model's coverage in our comparison. Higher coverage means that label model can produce more pseudolabels, yielding a larger size of training dataset to use.

**Expected Results.** We expect that synthesized programs may offer some advantages in terms of efficiency and effectiveness compared to human-designed ones.

**Results.** Table 4 presents their comparison. By leveraging the knowledge and capabilities of language models, we find that generated programs offer several advantages, including better coverage (i.e., the ability to label more data points) and comparable, or even better, performance. Generated programs can reduce the need for laborious engineering, which can be time-consuming and often requires a tedious design process to fine-tune labeling logic, such as thresholds and keyword usage. This design process may lead to many undiscovered rules, resulting in lower performance on coverage and potentially limiting the effectiveness of the labeling functions—unlike synthesized programs.

This is particularly evident in the SMS dataset, where WRENCH requires 73 manually crafted labeling functions to obtain high-quality labels, while Alchemist only needs 10 generated programs to obtain comparable performance and higher coverage. This significant reduction highlights the potential of Alchemist to *assist humans in designing labeling functions and make it more accessible to users without extensive domain expertise.*

# 5  Conclusion

We propose an alternative approach to costly annotation procedures that require repeated API requests for labels. Our solution introduces a simple notion of prompting programs to serve as annotators. We developed an automated labeling system called Alchemist to embody this idea. Empirically, our results indicate that Alchemist demonstrates comparable or even superior performance compared to language model-based annotation, improving five out of eight datasets with an average enhancement of 12.9%. Notably, Alchemist reduces total costs by a factor of approximately 500. Furthermore, we showcase the system's extensibility to handle more complex modalities while enhancing the robustness of predicted labels. Finally, we confirm that incorporating relevant information can generate better programs, and increasing diversity leads to obtaining higher-quality labels.

# 6  Acknowledgments

We are grateful for the support of the NSF under CCF2106707 (Program Synthesis for Weak Supervision) and the Wisconsin Alumni Research Foundation (WARF). We thank Dyah Adila, Albert Gu, Harit Vishwakarma, and Nicholas Roberts, for their helpful feedback and valuable discussion.

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

The appendix is organized as follows. First, we provide details about datasets, training settings, and computation resources in Appendix A. Next, in Appendix B we list prompts that we use to query language models. Then, we present ablation studies in Appendix C using other models in weak supervision to work with Alchemist. We include two additional modalities in Appendix D. We incorporate other work [51] for enhancing robustness into Alchemist and present in E. Lastly, we discuss limitations and broader impacts of our work in Appendix F.

## A   Datasets and Implementation Details

| Dataset | Task Type | Prediction Classes | # of Classes | # of Train |
|---|---|---|---|---|
| YouTube [43] | spam comment detection | {"spam", "ham"} | 2 | 1686 |
| SMS [44] | spam text detection | {"spam", "ham"} | 2 | 4571 |
| Yelp [45] | restaurant review sentiment classification | {"postive", "negative"} | 2 | 30400 |
| IMDb [45] | movie review sentiment classification | {"postive", "negative"} | 2 | 20000 |
| MedAbs [48] | medical abstract topic classification | {"neoplasms", "digestive system diseases", "nervous system diseases", "cardiovascular diseases", "general pathological conditions"} | 5 | 10395 |
| Cancer [14] | biomedical document topic classification | {"colon cancer", "lung cancer", "thyroid cancer"} | 3 | 5450 |
| Finance [46] | finance news sentiment classification | {"positive", "neutral", "negative"} | 3 | 3488 |
| French [47] | book review sentiment classification | {"positive", "neutral", "negative"} | 3 | 6953 |
| Waterbirds [42] | bird species classification | {"landbird", "waterbird"} | 2 | 5794 |

Table 5: Dataset Table.

We place more details about our datasets and experimental setups here. First, in Table 5 we show task type, prediction classes, and number of training data points in each dataset. MedAbs, Cancer, Finance, and French are considered to be more challenging settings, where these datasets typically need domain expertise to provide labels. Waterbirds is considered to test for a more complex modality.

We employ Snorkel as our label model to aggregate program outputs and report results in the main paper. We show more results using different choices of label model in Appendix C. All the distilled models use the MLP model that is trained with 2 hidden layers, each comprising 32 units, using ReLU activations between layers and no normalization. We run 5 times with different random seeds and report their average performance. We use a A6000 NVidia GPU to run all experiments.

## B   Used Prompts

| Dataset | Zero-shot Prompting (Baseline) |
|---|---|
| YouTube | what is the category of this youtube comment: [text] |
| SMS | what is the category of this sms text: [text] |
| Yelp | what is the sentiment of this restaurant review: [text] |
| IMDb | what is the sentiment of this movie review: [text] |
| MedAbs | what is the topic of this abstract: [text] |
| Cancer | what is the topic of this document: [text] |
| Finance | what is the sentiment of this news: [text] |
| French | what is the sentiment of this book review: [text] |

Table 6: Prompts for baseline approach are presented.

| Dataset | Task Description (Alchemist) |
|---|---|
| YouTube | Write a bug-free and executable function in python to label comment on Youtube as spam or ham. |
| SMS | Write a bug-free and executable function in python to label SMS text as spam or ham. |
| Yelp | Write a bug-free and executable function in python to label the sentiment of restaurant review on Yelp as postive or negative. |
| IMDb | Write a bug-free and executable function in python to label the sentiment of movie review on IMDB as postive or negative |
| MedAbs | Write a bug-free and executable function in python to label the topic of medical abstract. |
| Cancer | Write a bug-free and executable function in python to label the topic of biomedical document. |
| Finance | Write a bug-free and executable function in python to label the sentiment of financial news as postive, neutral, or negative |
| French | Write a bug-free and executable function in python to label the sentiment of book review written in French as postive, neutral, or negative. |

Table 7: Task descriptions in Alchemist's prompt are presented.

Next, we present the prompts used to query language models in the baselines and Alchemist.

|  | Youtube | | SMS | | Yelp | | IMDB | |
|---|---|---|---|---|---|---|---|---|
|  | Est. Cost | Accuracy | Est. Cost | F1-score | Est. Cost | Accuracy | Est. Cost | Accuracy |
| Zero-shot Prompting | 0.096 | 0.871 | 0.240 | 0.907 | 3.873 | **0.845** | 3.400 | **0.737** |
| Weighted Majority Vote | 0.004 | **0.874** | 0.004 | 0.886 | 0.005 | 0.705 | 0.004 | 0.520 |
| Dawid-Skene | 0.004 | 0.864 | 0.004 | 0.895 | 0.005 | 0.682 | 0.004 | 0.507 |
| FlyingSquid | 0.004 | 0.863 | 0.004 | **0.915** | 0.005 | 0.678 | 0.004 | 0.500 |
| Snorkel | 0.004 | **0.891** | 0.004 | 0.900 | 0.005 | 0.575 | 0.004 | 0.662 |
|  | MedAbs | | Cancer | | Finance | | French | |
|  | Est. Cost | Accuracy | Est. Cost | Accuracy | Est. Cost | Accuracy | Est. Cost | Accuracy |
| Zero-shot Prompting | 1.944 | 0.311 | 15.925 | 0.716 | 0.201 | 0.641 | 0.641 | 0.611 |
| Weighted Majority Vote | 0.006 | **0.354** | 0.003 | **0.968** | 0.007 | **0.650** | 0.006 | 0.221 |
| Dawid-Skene | 0.006 | 0.262 | 0.003 | **0.957** | 0.007 | **0.661** | 0.006 | 0.221 |
| FlyingSquid | 0.006 | **0.323** | 0.003 | **0.967** | 0.007 | **0.661** | 0.006 | **0.690** |
| Snorkel | 0.006 | **0.346** | 0.003 | **0.968** | 0.007 | **0.660** | 0.006 | **0.690** |

Table 8: Testing performance of the distilled model is reported for each combination of label model and dataset.

First, we show the prompts used for the baseline approach of zero-shot prompting on text datasets in Table 6. In these prompts, the placeholder "[text]" is replaced with individual data points and sent via API calls to obtain labels for each data point.

Next, we present the prompts used in Alchemist in Table 7. The table displays the task description component of each prompt. These descriptions outline the objective of the generated program and are associated with the prediction classes. For the labeling instructions, we directly map the prediction classes to their corresponding class indices and query the language models to output the appropriate class index.

For the image task, we use the prompts ["an image of landbird", "an image of waterbird"] to perform zero-shot prompting using CLIP. In Alchemist, we first query high-level concepts and then combine them with computed scores to prompt LLMs to generate programs. The first step involves the following prompt: "What are the visual primitive concepts to classify "landbird" and "waterbird"? Please organize the primitive concepts by name and use comparisons for the classes. Parse the results into JSON format."

Once we have obtained a set of similarity scores, we use the following prompt: "I have measured similarity scores for the following descriptions as float numbers. If a score is close to 1, it is highly related to the description. If a score is close to 0, it is less related to the description. The descriptions are: ["A bird's foot type is toed, grasping"]; ["A bird's foot type is paddling, swimming"]. Generate a labeling function with input scores to classify landbirds and waterbirds. If it cannot be determined, the function should return -1, but use this cautiously." Descriptions will be replaced by different generated concepts.

## C  Ablation Studies

Alchemist is compatible with a variety of weak supervision aggregation approaches. We report additional results with different choices of label models. Besides Snorkel, we consider three more widely-used label models: Weighted Majority Vote, Dawid-Skene [52], and FlyingSquid (FS) [28]. We reuse our experimental setup from Sec. 4.1 and in Sec. 4.5 and present the performance of the distilled models in Table 8 and in Table 9, respectively.

In Table 8, we observe that the label accuracy is enhanced or achieves comparable performance with different label models, showcasing Alchemist's flexibility in working with various label models. In Table 9, we include compare them with human-crafted labeling functions developed in WRENCH [50]. Similarly, Alchemist obtains higher coverage and achieves comparable or even better label accuracy while reducing the need to craft a large number of programs manually.

Next, we conduct an experiment by varying the temperature in our query APIs and running Alchemist on four different datasets. We train the end model five times with different random seeds and computed the average performance and the variance. Results are shown in Table 10. We observe

|  |  | Number of Programs | Coverage | Weighted Majority Vote | Dawid-Skene | FlyingSquid | Snorkel |
|---|---|---|---|---|---|---|---|
| Youtube | Human-crafted | 10 | 0.89 | 0.88 | 0.84 | 0.87 | 0.85 |
|  | GPT-3.5 | 10 | 1.00 | 0.87 | 0.86 | 0.86 | **0.89** |
|  | GPT-4 | 10 | 1.00 | 0.85 | **0.88** | **0.87** | **0.89** |
|  | Claude 3 | 10 | 1.00 | 0.77 | 0.71 | 0.73 | 0.72 |
| SMS | Human-crafted | 73 | 0.41 | 0.90 | 0.86 | 0.00 | 0.89 |
|  | GPT-3.5 | 10 | 1.00 | 0.89 | 0.90 | 0.90 | 0.90 |
|  | GPT-4 | 10 | 1.00 | **0.91** | 0.90 | **0.92** | **0.93** |
|  | Claude 3 | 10 | 1.00 | **0.91** | **0.92** | **0.92** | 0.89 |
| Yelp | Human-crafted | 8 | 0.83 | 0.75 | 0.83 | 0.77 | 0.76 |
|  | GPT-3.5 | 10 | 0.78 | 0.70 | 0.68 | 0.68 | 0.57 |
|  | GPT-4 | 10 | 0.99 | 0.73 | **0.81** | 0.72 | 0.82 |
|  | Claude 3 | 10 | 0.88 | **0.77** | 0.78 | **0.81** | **0.83** |
| IMDb | Human-crafted | 5 | 0.88 | 0.72 | 0.73 | 0.68 | 0.73 |
|  | GPT-3.5 | 10 | 0.89 | 0.52 | 0.51 | 0.50 | 0.66 |
|  | GPT-4 | 10 | 1.00 | 0.54 | 0.55 | 0.54 | **0.75** |
|  | Claude 3 | 10 | 0.98 | 0.59 | 0.64 | 0.60 | 0.70 |

Table 9: We offer a comparison between a wider range of label model options for synthesized programs and those designed by humans.

| | | YouTube | | | | SMS | | | | French | | | | Cancer | | | |
|---|---|---|---|---|---|---|---|---|---|---|---|---|---|---|---|---|---|
| | | Label Model | | End Model | | Label Model | | End Model | | Label Model | | End Model | | Label Model | | End Model | |
| | | Mean | Std. | Mean | Std. | Mean | Std. | Mean | Std. | Mean | Std. | Mean | Std. | Mean | Std. | Mean | Std. |
| | 0.0 | 0.755 | 0.020 | 0.795 | 0.005 | 0.589 | 0.000 | 0.866 | 0.014 | 0.550 | 0.001 | 0.690 | 0.000 | 0.729 | 0.000 | 0.935 | 0.009 |
| Temperature | 0.5 | 0.898 | 0.002 | 0.870 | 0.005 | 0.603 | 0.013 | 0.854 | 0.019 | 0.519 | 0.000 | 0.690 | 0.001 | 0.733 | 0.000 | 0.940 | 0.010 |
| | 1.0 | 0.817 | 0.004 | 0.803 | 0.017 | 0.667 | 0.022 | 0.930 | 0.012 | 0.555 | 0.001 | 0.690 | 0.000 | 0.731 | 0.000 | 0.938 | 0.006 |

Table 10: We varied the temperature in GPT-4 API calls, showing **consistent performance** across four datasets. We also trained the end model five times, displaying the average performance and the variance. These results confirm the stability of Alchemist.

consistent labeling performance across different temperatures (0.0, 0.5, and 1.0), demonstrating Alchemist's stability. Additionally, the stability of generated programs highlights the significance of including aggregation models to handle noisy and diverse outputs, resolve conflicts, and produce final labels.

# D  Richer Modalities

We run Alchemist in two additional modalities in Table 11: time-series (ECG heartbeat classification [53]) and tabular (Census income classification [54]). For ECG heartbeat classification, we generated 10 labeling programs from GPT-4o. For the Census income dataset, we generated program codes for each attribute (e.g., gender, education, age, race). We used Snorkel as our label model. The results demonstrate Alchemist's capability to handle more complex modalities and produce satisfactory performance. We compared Alchemist with human-crafted labeling functions from WRENCH. Alchemist uses fewer labeling functions (programs) and reaches higher labeling performance.

In general, Alchemist will work well with any of these modalities as long as we have access to any cheap local feature extractor. This includes medical imaging tasks: [55] showed manually-crafted simple labeling functions were able to identify heart problems in MRI sequences based on very simple primitives, which could act as the feature extractors for Alchemist.

# E  Improving Robustness

We integrate advancements from existing robustness techniques into Alchemist to further improve accuracy and reduce spurious correlations. Specifically, we consider RoboShot [51], which prompts LLMs for spurious and correct correlation features, then calibrates image embeddings by projecting them to reject or accept concepts. In our setup, we use GPT-4o to identify spurious correlations for

|  | ECG Heartbeat Classification | | Census Income Classification | |
|---|---|---|---|---|
|  | Human Crafted | Synthesized by GPT4o | Human Crafted | Synthesized by GPT4o |
| Modality | **Time-Series** | | **Tabular** | |
| # of Train Set / # of Test Set | 87554 / 21892 | | 30162 / 15060 | |
| # of Classes | 5 | | 2 | |
| # of Programs | — | 10 | 83 | **13** |
| Label Model Accuracy | — | **0.827** | 0.681 | **0.725** |

Table 11: We included two new modalities in our evaluation: **time-series and tabular data.** Both performed well using a few generated programs. In the Census dataset, Alchemist outperformed human-crafted labeling functions in WRENCH.

| Feature Extractor | Method | Average Accuracy (↑) | Worst Group Accuracy (↑) | Gap (↓) |
|---|---|---|---|---|
| CLIP ViT-B/32 | Zero-shot Prompting | 0.820 | 0.318 | 0.502 |
|  | Group Prompting | 0.823 | 0.383 | 0.440 |
|  | **Alchemist with GPT4o** | 0.805 | 0.283 | 0.522 |
|  | **RoboShot + Alchemist with GPT4o** | 0.700 | **0.375** | **0.325** |
| CLIP ViT-L/14 | Zero-shot Prompting | 0.904 | 0.335 | 0.569 |
|  | Group Prompting | 0.791 | 0.240 | 0.551 |
|  | **Alchemist with GPT4o** | 0.802 | 0.467 | 0.335 |
|  | **RoboShot + Alchemist with GPT4o** | **0.803** | **0.569** | **0.234** |

Table 12: We integrate Roboshot into Alchemist by querying GPT-4 for spurious correlation features and rejecting them, then reusing the generated programs in Alchemist. This integration improves average accuracy and worst-group performance, **enhancing robustness to spurious correlations.**

classifying waterbirds and landbirds, then project image embeddings onto these concept embeddings to reject spurious correlations through subtraction. We compute cosine similarity using the calibrated embeddings to obtain score sets, which are then fed into Alchemist's generated programs. The spurious correlations identified by GPT-4o are ["water background", "land background", "aquatic plants", "trees and bushes"]. Results are displayed in Table 12. This integration using GPT-4o successfully enhances robustness to spurious correlations by improving worst-group accuracies.

# F    Discussion

**Techniques to Evaluate Generated Programs.**    There are many ways to evaluate the quality of generated programs in advance. Expert users can quickly determine whether key problem properties are being used by looking at the code. Besides human inspection, Alchemist includes several automated measurement tools to diagnose generated programs. First, we analyze program outputs to compute coverage, polarity, conflict, and overlap (see [3] for definitions). For example, if coverage (the fraction of data points with at least one label) is below 10%, we discard the program and ask users to generate a new one. Moreover, if a validation dataset is available, Alchemist can run diagnostics to empirically compare accuracy with ground truth, offering more insight into the program's reliability. This data-driven feedback loop ensures tractable program generation. Notably, these tools are not typically accessible with model-based annotation methods.

**Limitations.**    There are two primary limitations in Alchemist. First, the performance of the datasets we test is still dependent on the capabilities of the language model. If the language model's ability to comprehend the given task and generate effective programs is subpar, the labeling performance may suffer. The second limitation arises when dealing with extremely complex tasks. As the complexity of the task increases, the generated code may become longer, more intricate, and harder to understand, posing challenges for developers who take time to validate correctness.

---

[3]https://snorkel.readthedocs.io/en/v0.9.3/packages/_autosummary/labeling/snorkel.labeling.LFAnalysis.html

**Broader Impacts.** We do not see explicit negative impacts in Alchemist's annotation process. However, generated programs from language models may contain biased labeling logic, toxic content, or malicious functions. To mitigate this, auditing and guardrails may be necessary.

