# OpenReview forum: "The ALCHEmist: Automated Labeling 500x CHEaper than LLM Data Annotators"
_NeurIPS.cc/2024/Conference — NeurIPS 2024 spotlight_

### Official Review · Reviewer_b4or · 2024-07-03

**Soundness:** 3
**Presentation:** 3
**Contribution:** 3
**Rating:** 5
**Confidence:** 4

**Summary:**

The paper proposes a labeling workflow (Alchemist) using LLMs where instead of labeling each data point using a teacher model, we ask the teacher model to generate a program to label for the given task. Multiple such programs are generated and using an aggregation function, we get pseudo labels. These pseudo labeled points are then used as training data to train a student model. They show that such labeling mechanism is effective and less expensive compared to the teacher model labeling every data point. They also perform ablation studies for multi-modality, supplementary information during program generation, and diversity of programs.

**Strengths:**

1. Knowledge distillation using a larger LLM to label datasets is very prevalent now. It is an expensive process.
2. Alchemist is an interesting way to reduce costs.
3. The idea to generate weak labeling functions/programs is novel and as the weak supervision literature shows, it can be effective in certain cases.
4. The paper is clearly written, and the experiments section is organized to answer critical questions.

**Weaknesses:**

As the results show, no performance degradation is not guaranteed (for e.g. table 1, Yelp dataset). For a new dataset, without comparing the approaches, it may be hard to understand if Alchemist works for it without impacting the performance.

**Questions:**

1. Why are the accuracy numbers for Alchemist with GPT-3.5 different in table 1 and table 3?
2. How does the performance compare if we use the generated programs for labeling the test set instead of training the student model?
3. Given that the performance gain/no loss is not consistent across datasets, what is the recommendation for a new dataset, i.e. how can one know whether this technique is going to work for that dataset? There are mentions that it works better for complex dataset, any other detailed characterization possible?

**Limitations:**

The main limitation is that it is hard to understand if Alchemist is going to work for your task without spending the money to label and compare. Performance loss when observed is not small to ignore. Generated programs may be reviewed by human experts, but they may not be straight-forward to interpret and gain confidence from.

---

> ### Author Rebuttal · Authors · 2024-08-06
>
> ### Response to Reviewer b4or
>
> We are grateful for the review and the positive assessment. Thank you for acknowledging Alchemist is novel, interesting, and our paper is well-written. We address your questions below.
>
> * **On Performance Degradation.**
>     * In the majority of cases (six out of eight datasets), Alchemist **performs as well as or even better than model-based annotation, in addition to being orders of magnitude cheaper, more extensible, and easier to inspect**. In fact, our results suggest that users should prefer Alchemist over model-based annotation.
>     * In cases where a user seeks to improve Alchemist's base performance, we can simply add more generated programs---**at extremely low cost.** In Table 1 in our paper, we report results with 10 generated programs. Increasing the number of generated programs enhances performance, as we showed in Figure 5, where generating 20 labeling programs for Yelp and IMDb datasets results in **better labeling performance compared to zero-shot prompting.**
> * **On Accuracy Performance in Table 1 and Table 3.**
>     * Results shown in Table 1 are for a model trained on Alchemist-created labels (we report performance on the testing dataset), while in Table 3, we present the label model's performance (i.e., just using the ensemble of programs that Alchemist generates) on the testing dataset. These two approaches correspond to *two stages in the weak supervision process*: first, create a label model that just uses the labeling functions (programs, in our case), and second, obtain labels from the label model and train an *end model*. Using just the label model and training an end model have both been extensively studied in the weak supervision literature. It is known that training an end model on weakly-labeled data enables generalizing beyond the label model in many cases. The reason for this is based on the observation that the label model may not be capable of capturing all labeling patterns (i.e., all possible features). It may instead only use certain simple ones. Training an end model permits learning many patterns, both simple and complex, which enables better generalization in many scenarios. **Our results are consistent with these notions.**
> * **On Labeling the Test Set.**
>     * We described the difference between the label model and the end model above. We organized Alchemist's results in Table 5 in our attachment. We show the testing performance using the label model and the trained end model for eight datasets. **In most of the cases, the end model is preferable to the label model.**
> * **On Techniques to Evaluate Generated Programs.**
>     * There are many ways to evaluate the quality of generated programs in advance. First, expert users can quickly determine whether key problem properties are being used by looking at the code. Besides human inspection, Alchemist includes several automated measurement tools to diagnose generated programs. We analyze program outputs and compute their coverage, polarity, conflict, and overlap (see [1] for definitions). For instance, coverage measures the fraction of data points with at least one label. If coverage is below a certain bar (e.g., 10%), we discard the program and ask users to generate a new one.
>     * Moreover, if a validation dataset is available, Alchemist can run diagnostics to empirically compare accuracy with ground truth, offering more insight into the program's reliability. **Notably, these tools are not typically accessible with model-based annotation methods.**
> * **On Data Characteristics.**
>     * In general, we have found that Alchemist works well when the labeling logic involves a mixture of logical formulas over a set of salient primitives. This captures most cases of interest. Formalizing a definition of what data or task characteristics are relevant is a very interesting problem for future work; we foresee studying, for example, the expressivity of programs and understanding if they can meet most ML tasks. Kolmogorov complexity or similar notions will likely be helpful here.
> * **On Framework Limitation.**
>     * This limitation is common across many annotation approaches. We often only know the results after completing the entire labeling process, which is also true for model-based annotation methods. However, Alchemist significantly reduces expenses, requiring a lower order of magnitude and enabling more efficient reruns of the labeling process. In other words, **Alchemist offers a much more tractable approach to iteration compared to model-based annotation**.
>
> [1] https://snorkel.readthedocs.io/en/v0.9.3/packages/_autosummary/labeling/snorkel.labeling.LFAnalysis.html

---

> > ### Comment · Reviewer_b4or · 2024-08-12
> >
> > I thank the authors for the detailed responses to my questions and additional experiments.

---

> > > ### Author Response · Authors · 2024-08-12
> > >
> > > Thank you! Your suggestions helped us to improve the paper. Please let us know if you have any further questions; we are very happy to follow up. If there are no further concerns, we would appreciate it if you increase your score---we appreciate it!

---

### Official Review · Reviewer_d5ZD · 2024-07-12

**Soundness:** 4
**Presentation:** 3
**Contribution:** 3
**Rating:** 6
**Confidence:** 4

**Summary:**

The authors propose an innovative solution for high cost of APIs that using large pretrained models to generate programs that act as annotators. This idea helps replace or supplement crowdworkers and allows for distilling large models into smaller, more specialized ones. Traditional methods can be costly and produce static, hard-to-audit datasets due to expensive API calls. To tackle this, they introduce Alchemist, a system that tasks models to generate reusable labeling programs. These programs can be applied locally, significantly cutting costs while maintaining or even improving performance.

**Strengths:**

1. Generating programs that can produce labels is an interesting idea. This paper introduces a mechanism that can generate multiple labels, transforming the one-to-one mapping between samples and labels into a sustainable one-to-many or many-to-many relationship. This approach can significantly reduce API costs.

2. From a practical implementation standpoint, Alchemist's flexibility and reusability are major strengths. The system allows users to create simple prompts for generating labeling programs that can be stored, reused, and extended. This adaptability makes it a versatile tool suitable for a wide range of tasks and ensures that users can tailor it to their specific needs.

2. The code is straightforward and easy to read, making it accessible even for those who might not be deeply familiar with the underlying concepts.

**Weaknesses:**

1. One of the limitations of Alchemist is that the generated programs tend to handle only fixed tasks, often relying on threshold-based judgments. This means that for more complex or varied tasks, the system might still need to call specialist models via API, which somewhat limits its flexibility.

2. Another weakness is the lack of experimental evidence regarding the stability of the generated programs. For instance, the paper doesn't thoroughly explore how factors like the temperature variable in the code could affect the quality and consistency of the generated labeling programs.

3. The authors could also improve their literature review. There are several highly relevant papers that discuss various aspects of data annotation that weren't cited, such as [1] [2] [3] [4] [5].

[1] https://arxiv.org/abs/2310.04668
[2] https://arxiv.org/abs/2303.15056
[3] https://arxiv.org/abs/2306.04349
[4] https://dl.acm.org/doi/pdf/10.1145/3613904.3642834
[5] https://dl.acm.org/doi/pdf/10.1145/3594536.3595161

**Questions:**

Refer to weaknesses.

---

> ### Author Rebuttal · Authors · 2024-08-06
>
> ### Response to Reviewer d5ZD
>
> Thank you for recognizing Alchemist’s flexibility, extensibility, and for your kind words about our paper! We address your questions below and include experimental results on more complex tasks and program stability.
>
> * **On More Complex Tasks.**
>     * Alchemist is not limited to threshold-based simple tasks. It is capable of capturing more complex labeling logic, as we have observed in some of the generated programs. To further demonstrate this capability, we included two more challenging datasets: **ECG heartbeat classification** and **Census income classification.** Results are shown in Table 2 in our attachment and demonstrate that Alchemist can handle more complex modalities and perform well. Moreover, we compared Alchemist with human-crafted labeling functions from WRENCH. **Alchemist uses fewer labeling functions (programs) and reaches higher labeling performance.**
>     * Additionally, since submission, we have been working on an additional, highly timely, application of this work to LLM-as-a-judge. This is a crucial area since users and companies spend vast sums of money for evaluating model outputs using automated LLM-as-a-judge techniques. We have collaborated with a leading startup in the industry to use our technique in production environments to reduce costs. Our preliminary results in this area suggest that Alchemist can capture sufficiently complex logic to identify poor-quality responses from language model pipelines.
> * **On the Stability of Generated Programs.**
>     * This is a great question. We conducted an experiment by varying the temperature in our query APIs and running Alchemist on four different datasets. We trained the end model five times with different random seeds and computed the average performance and the variance. Results are shown in Table 4 in our attachment. We observe **consistent labeling performance** across different temperatures (0.0, 0.5, and 1.0), demonstrating Alchemist’s stability. Additionally, the stability of generated programs highlights **the significance of including aggregation models** to handle noisy and diverse outputs, resolve conflicts, and produce final labels.
> * **On Improving Literature Review.**
>     * Thank you for sharing these papers with us. We have included them in our related work section. **Key differences**: These papers prompt LLMs for labels **per-sample**, requiring a large number of API calls (scaling with the size of the dataset) and making it hard to inspect mistakes and revise labeling logic, unlike Alchemist.

---

> > ### Comment · Reviewer_d5ZD · 2024-08-13
> >
> > The authors have satisfactorily addressed most of my problems. Most concerns has been addressed and some senarios may out of scope of this paper. I have raised my score. Once again, I want to express my gratitude for your hard work and commitment.

---

> > > ### Author Response · Authors · 2024-08-13
> > >
> > > Thank you for your response and valuable feedback. Your suggestions have improved our paper. If you have any further questions, feel free to ask—we’re happy to provide more information. Thanks again!

---

### Official Review · Reviewer_uwkF · 2024-07-12

**Soundness:** 3
**Presentation:** 4
**Contribution:** 4
**Rating:** 8
**Confidence:** 4

**Summary:**

The paper presents a new method for creating data labels that leverage a Large Language Model and the weak supervision/data programming labeling paradigm. In this work, the LLM is used to generate labeling code, typically in the form of functions, which can then be used to create weak labels for weak supervision. The paper goes on to show this method’s utility across several benchmark text datasets and one image dataset. The paper also does an ablation study showing the impact of different elements of the method, like numbers of weak functions, and including examples in the prompt to create the functions.

**Strengths:**

The paper is strong in its significance, clarity, and quality. For significance, the paper is attacking a significant problem. One of the great promises of large ML models, with multimodal or text only, is their ability to zero-shot label text. This ability allows LLMs to possibly overcome one of – if not the chief issue – with building machine learning models: labeled data. Thus, this paper is attacking a profoundly important problem for the application of ML to real-world problems. For clarity,  the paper is well-written and the diagrams are very helpful. When combined with the appendices and supplementary material, I can not only easily see how to reproduce their results, but how to apply this method to my data labeling problems. In other words, I can easily see wide adoption and use of this method. Finally, the paper does a reasonably good job in its empirical testing to cover many variations on the application of the method (e.g., having both text and image datasets) and variations to the method (e.g., including examples, numbers of weak functions, etc.).

**Weaknesses:**

The weakness of the paper is in its grounding/novelty and some of its methods, particularly when applied to other modalities. For the grounding, while the paper actually does capture many of the previous works that have done something similar, there are works like Cruickshank and Ng, “DIVERSE: Deciphering Internet Views on the U.S. Military Through Video Comment Stance Analysis, A Novel Benchmark Dataset for Stance Classification” where they used LLMs + weak supervision to create an actual new dataset. They did not, however, have the novel insight about creating the weak labeling functions that this work does.

For the method in the image or multimodal case, two works have done something very similar with CLIP. First, Adila et al. “Zero-shot Robustification of Zero-shot Models” use the same CLIP model and water birds dataset, but get substantial performance increases by “debiasing” the image embeddings, with text characteristics. This is very similar to what is done with the labeling functions where this paper tries to get labeling functions to classify aspects of the birds and ignore spurious contexts. Second, Bendou et al.’s “LLM meets Vision-Language Models for Zero-Shot One-Class Classification” presents a method for one-shot classification using VLMs, which uses an LLM to build negative classes around a positive class. This is similar to the insights in this paper around developing labeling functions to highlight important visual distinctions in the classes to improve the labels. Taken together, I think this paper might be able to incorporate the insights from these other papers to actually improve their results. For example, the debiasing in the first paper could be used with the proposed method.

**Questions:**

I have no additional questions.

**Limitations:**

One other societal limitation the work could address is how works like this are changing the data labeling industry. Data labeling is still a massive, multi-million-dollar industry and these works are changing the nature of that industry. Ideally, they are changing it positively, but it can still disrupt how the data labeling industry works and cost human labelers their income.

---

> ### Author Rebuttal · Authors · 2024-08-06
>
> ### Response to Reviewer uwkF
> We are grateful for your review and for describing our paper as significant, well-written, and of high quality. We address your questions below and provide additional experimental results incorporating Roboshot into Alchemist!
>
> * **On Grounding.**
>     * Thank you for sharing this paper with us. We agree it relates to Alchemist, and we have included it in our updated related work. **Key differences** include: unlike Alchemist, which automatically generates programs, this work requires domain expertise and more human effort to craft heuristics and search for proper keywords.
>     * In Sec. 4.4 in our paper, we include an experiment comparing Alchemist to human-crafted labeling functions. For example, in the SMS dataset, WRENCH requires 73 manually crafted labeling functions to obtain high-quality labels, whereas **Alchemist only needs 10 generated programs to achieve comparable performance and higher coverage**. This significant reduction highlights Alchemist's capability to assist humans in generating label sources, making it more accessible to users without extensive domain expertise.
> * **On Two Similar Works.**
>     * Thank you for pointing out these two papers. The first paper prompts LLMs for spurious and correct correlation features, then calibrates image embeddings by projecting them to reject or accept concepts. The second paper asks LLMs for confusing visual objects to perform zero-shot one-class classification with CLIP. We have included them in the draft.
> * **On Incorporating Roboshot into Alchemist.**
>     * This is an interesting idea. We implemented this suggestion by incorporating Roboshot into Alchemist. We first used GPT-4o to identify spurious correlations for classifying waterbirds and landbirds, then projected image embeddings onto these concept embeddings to reject spurious correlations through subtraction. We computed cosine similarity using the calibrated embeddings to obtain score sets, which were then fed into Alchemist’s generated programs. The spurious correlations identified by GPT-4o were *{'water background,' 'land background', 'aquatic plants,' 'trees and bushes'}*. Results are displayed in Table 3 in the attachment. We observe that integrating Roboshot with Alchemist using GPT-4o **enhanced robustness to spurious correlations by improving accuracies.**
> * **On Framework Limitations.**
>     * Our motivation is to address the **drawbacks of pretrained model-based annotation, which is expensive, lacks extensibility, and makes results hard to inspect**. We believe Alchemist serves as a better labeling tool by reducing costs and providing human labelers with more power to inspect and refine labeling results. In addition, generated programs can serve as templates for human labelers to rewrite, extend, or customize labeling logic. These benefits make the labeling process easier and more efficient for humans.

---

> > ### Comment · Reviewer_uwkF · 2024-08-13
> > **Reply to Rebuttal**
> >
> > The Authors have improved the paper further. I am particularly impressed to see the Roboshot idea incorporated and that it further enhanced the method. I stand by my rating of this being string accept.

---

> > > ### Author Response · Authors · 2024-08-13
> > >
> > > Dear reviewer,
> > >
> > > Thank you for your valuable suggestion and your support. We are excited about the new results with Roboshot as well. We appreciate it! Thank you for your time!

---

### Official Review · Reviewer_n36E · 2024-07-12

**Soundness:** 3
**Presentation:** 3
**Contribution:** 3
**Rating:** 7
**Confidence:** 5

**Summary:**

The paper proposes an automated way to label large quantities of data by leveraging large language models to generate labeling functions which can be used to label data using programmatic weak supervision. The paper demonstrates that this procedure can generate labeling functions which are more accurate than manually generated LFs and querying LLMs for labels directly at a fraction of cost. The authors also show that prompt tuning benefits their method, and that it can be used for other richer modalities.

**Strengths:**

1. The paper is well written, and easy to follow.
2. The experiments are well designed.
3. The idea is simple, intuitive and has promising performance.
4. Evaluation on richer data modalities is challenging, and I appreciate the authors' inclusion of image datasets.

**Weaknesses:**

Most of these concerns are minor, and some of them can be readily fixed.
1. **Comparisons with related work:**: I believe that comparisons with empirical ScriptoriumWS and DataSculpt are necessary to demonstrate the benefits of the proposed approach. With that said, I found the experiments to be rigorous. I would also like the authors to catalog the differences between their approach and ScriptoriumWS, DataSculpt, and [1] to better highlight the novelty of their methodology and findings. I am aware that [1] was published very close to the NeurIPS deadline so I do not expect the authors to compare their methods with [1], but conceptual differences (if any) can still be highlighted.
2. **Missing references:**: The idea of using keywords and LLMs to automatically create LFs is not new, so I would encourage the authors to reference some prior work (e.g. [2]).
3. **Lack of reproduciblity and claims:** The techniques presented in the paper and the results are not reproducibile with the details given in the paper. I would encourage the authors to release their code for the same. The paper mentions that "any particular program may be inaccurate, fail to compile, or may otherwise be flawed, ...". I am curious how the authors ensure that the programs are not downright inaccurate (because snorkel would require the accuracy of LFs to be greater than random chance), or fail to compile? I believe that some kind of a data-driven feedback loop is important to ensure that the programs compile, and the generated LFs have coverage, and are accurate. Therefore, I am excited about the possibility to include keywords, dataset descriptions, example in the prompt, but I feel that there has to be a feedback loop after the label model labels the data or identifies inaccurate LFs.
4. **Richer modalities:** The claim on scaling to richer modalities should be softened, because the authors only evaluate their approach on a relatively simple image dataset. More challenging tasks may include labeling chest X-rays [3], datasets for which are publicly available, and other richer modalities such as time series data [4]. I do not expect the authors to conduct these experiments during the rebuttal, but such an extension would be very interesting and valuable, in my opinion.

### References
1. Smith, Ryan, et al. "Language models in the loop: Incorporating prompting into weak supervision." ACM/JMS Journal of Data Science 1.2 (2024): 1-30.
2. Gao, Chufan, et al. "Classifying unstructured clinical notes via automatic weak supervision." Machine Learning for Healthcare Conference. PMLR, 2022.
3. Irvin, Jeremy, et al. "Chexpert: A large chest radiograph dataset with uncertainty labels and expert comparison." Proceedings of the AAAI conference on artificial intelligence. Vol. 33. No. 01. 2019.
4. Goswami, Mononito, Benedikt Boecking, and Artur Dubrawski. "Weak supervision for affordable modeling of electrocardiogram data." AMIA Annual Symposium Proceedings. Vol. 2021. American Medical Informatics Association, 2021.

**Questions:**

I wonder what the authors thoughts are on points 3 and 4 above in the weaknesses.

**Limitations:**

The authors have discussed the limitation of their approach.

---

> ### Author Rebuttal · Authors · 2024-08-06
>
> ### Response to Reviewer n36E
>
> Thank you for recognizing our paper as well-written with well-designed experiments. We appreciate your acknowledgment of our simple, effective idea and the significance of including diverse modalities. We appreciate your thoughtful review!
>
> * **On Comparisons with Related Work.**
>     * We compared Alchemist with related works in Table 1 in our attachment. **Alchemist uses fewer programs while achieving comparable labeling accuracy.** **Key differences** include:
>         * **Motivation:** Both ScriptoriumWS and DataSculpt address weaknesses in weak supervision, whereas Alchemist focuses on the downsides of model-based annotation.
>         * **Approach:** In ScriptoriumWS, supplementary information is manually crafted into prompts to generate code, while Alchemist uses LLMs to distill self-knowledge, enhancing labeling performance but using less human effort. DataSculpt employs *hundreds of programs for obtaining high-quality labels, but Alchemist achieves comparable accuracy with just 10 programs*.
>         * **Complex Modalities:** Alchemist handles complex modalities beyond text, while ScriptoriumWS and DataSculpt *do not*.
>     * PromptWS significantly differs from Alchemist. PromptWS prompts LLMs *multiple times for each sample*, **resulting in more API calls (even compared to zero-shot prompting).** While this method can improve performance, it incurs extremely high labeling costs, is harder to audit, and lacks extensibility. Alchemist, in contrast, addresses these issues.
> * **On Suggested References.**
>     * Thank you for sharing this paper. We have updated our related work and included it. While this work shares a similar idea with Alchemist, Alchemist goes beyond keyword extraction by capturing more complex and diverse labeling logic, and organizing them into executable programs (see Figure 1 in the paper). Moreover, extending to non-text modalities like image classification is challenging, but Alchemist is capable of this.
> * **On Reproducibility.**
>     * We attached our code in the supplementary material of our submission. We will release our code and a full set of outputs, including generated programs, after paper notification.
> * **On the Data-Driven Feedback Loop.**
>     * This is a good question, and it is a crucial step. Alchemist includes several tools for diagnosing generated programs. First, we analyze program outputs to compute coverage, polarity, conflict, and overlap (see [1] for definitions). For example, if coverage (the fraction of data points with at least one label) is below 10%, we discard the program and ask users to generate a new one. Second, if a validation dataset is provided, Alchemist computes empirical accuracy by comparing outputs with ground truth. Such a data-driven feedback loop ensures tractable program generation. We have included this step's description in the updated draft.
> * **On Richer Modalities.**
>     * Thank you for the suggestion! We included two additional modalities in Table 2: **time-series** (ECG heartbeat classification [2]) and **tabular** (Census income classification [3]). For ECG heartbeat classification, we generated 10 labeling programs from GPT-4o. For the Census income dataset, we generated program codes for each attribute (e.g., gender, education, age, race). We used Snorkel as our label model. The results demonstrate Alchemist's capability to handle more complex modalities and produce satisfactory performance. Moreover, we compared Alchemist with human-crafted labeling functions from WRENCH. Alchemist **uses fewer labeling functions (programs) and reaches higher labeling performance.** In general, Alchemist will work well with any of these modalities as long as we have access to any cheap local feature extractor. This includes medical imaging tasks: [4] showed manually-crafted simple labeling functions were able to identify heart problems in MRI sequences based on very simple primitives, which could act as the feature extractors for Alchemist.
>
> [1] https://snorkel.readthedocs.io/en/v0.9.3/packages/_autosummary/labeling/snorkel.labeling.LFAnalysis.html \
> [2] https://physionet.org/content/mitdb/1.0.0/ \
> [3] https://archive.ics.uci.edu/dataset/20/census+income \
> [4] Fries, J.A., Varma, P., Chen, V.S. et al. Weakly supervised classification of aortic valve malformations using unlabeled cardiac MRI sequences. Nat Commun 10, 3111 (2019).

---

> > ### Comment · Reviewer_n36E · 2024-08-12
> > **Thank you for the rebuttal!**
> >
> > Dear Authors,
> >
> > Thank you so much for your rebuttal and putting in time and effort. I have raised my score to reflect my current assessment of the paper.
> >
> > Below are some general thoughts based on the rebuttal:
> > 1. In addition to data-driven feedback, I also think that output driven feedback is important. In some cases the LLM might hallucinate and not generate LF code grounded on the input prompts, in which case an output-driven feedback loop can ideally guide the model to generate correct programs (accuracy is the second step, the model must generate correct programs first). I foresee this to be a problem in non-text modalities where the model is having to use APIs or local feature extractors. I think some discussion on this and your empirical findings would be beneficial to the community.
> > 2. "*Both ScriptoriumWS and DataSculpt address weaknesses in weak supervision, whereas Alchemist focuses on the downsides of model-based annotation.*" Can you explain this statement?
> > 3. It seems that ScriptoriumWS and Alchemist are similar in terms of accuracy and the # of programs which are generated. It seems from your description that the difference lies in extensibility to other richer modalities. Can you explain what exactly prevents the core ScriptoriumWS methodology to model richer modalities? The answer can be as simple as, "the method *can* be extended with X, Y and Z, but this was not evaluated".
> > 4. I would like to authors to provide some details about their time series classification experiment. The results are encouraging, and this is out of curiosity. I re-read the section on richer modalities in the paper, and the extension to time series doesn't seem trivial. I would like to know more about how the authors carried out this experiment.

---

> ### Author Response · Authors · 2024-08-13
>
> Dear reviewer,
>
> Thank you for your valuable feedback! Your suggestions have improved our paper. Appreciate it. We answer your questions below.
>
> 1. Thank you for your suggestion! We agree and have added additional descriptions to the paper addressing the output-driven feedback loop. One way to handle the challenge of making this work in non-text modality settings is to develop a set of simple unit tests that can be used to drive the feedback loop.
>
> 2. The difference comes down to what the goals of these techniques are. ScriptoriumWS and DataSculpt focus on automating weak source (labeling function) creation, addressing the limitations of human-crafted labeling functions, which require subject matter experts to implement by hand. The ideal outcome of these systems is to speed up the process of performing weak supervision by reducing the complexity of writing correct code. In contrast, Alchemist addresses the downsides of model-based annotation, such as high cost, lack of extensibility, and difficulty in auditing. Its goal is to efficiently distill model capabilities. While the mechanisms currently used to accomplish these goals have similarities, they are motivated by different challenges.
>
> 3. Indeed, ScriptoriumWS does not operate on non-text modalities at all; that technique is to directly prompt a language model to generate programs/labeling functions that perform a text task. The most straightforward way to extend the ScriptoriumWS methodology is to prompt a multimodal model like GPT4o to generate programs over the target modality (e.g., images). There are two downsides we have identified to this approach to extending ScriptoriumWS:
>     * It requires access to a powerful multimodal model, which may not exist for many modalities of interest,
>     * Even if such multimodal models exist, they may struggle with spurious correlations.
>
> While we did not directly evaluate ScriptoriumWS with GPT4o, we did extend Alchemist itself in this fashion as a baseline. In Table 2 in our paper, we reported our findings, which suggest that the spurious correlation issue is indeed a problem. In contrast, Alchemist’s approach, based on obtaining primitives and using a cheap feature extractor, reduces the impact of these spurious correlations. It additionally also mitigates the other challenge, as such feature extractors are much easier to obtain compared to a powerful multimodal model.
>
> 4. In this setting, we did not use the two-stage extension method to generate programs. Instead, we included supplementary information, such as dataset and prediction class definitions, directly in the prompts. Following the basic recipe for our work, we generate 10 programs and use Snorkel as the aggregation method to produce predictions. We show two generated programs below as examples. They use peak count as their labeling logic. The first program achieved 69.5% accuracy, while another reached 81.5%.
>
> ```
> def label_by_fft_peak(time_series):
>     """ Label based on the frequency domain characteristics """
>     fft_result = np.fft.fft(time_series)
>     fft_magnitude = np.abs(fft_result)
>     peak_freq = np.argmax(fft_magnitude[1:]) + 1  # Ignoring the DC component
>     if peak_freq < 15:
>         return 0  # 'N'
>     elif peak_freq < 30:
>         return 1  # 'S'
>     elif peak_freq < 45:
>         return 2  # 'V'
>     elif peak_freq < 60:
>         return 3  # 'F'
>     else:
>         return 4  # 'Q'
> ```
> ```
> def label_by_peak_count(time_series):
>     """ Label based on the number of peaks in the signal """
>     from scipy.signal import find_peaks
>     peaks, _ = find_peaks(time_series, height=0)
>     peak_count = len(peaks)
>     if peak_count > 7:
>         return 0  # 'N'
>     elif peak_count > 5:
>         return 1  # 'S'
>     elif peak_count > 3:
>         return 2  # 'V'
>     elif peak_count > 1:
>         return 3  # 'F'
>     else:
>         return 4  # 'Q'
> ```

---

### Author Rebuttal · Authors · 2024-08-06

### General Response
We are grateful for all the comments and constructive feedback on our work. Reviewers consistently found our paper to be well-written and easy to follow and described our work as novel and offering promising performance.

Reviewer **n36E** complimented the inclusion of complex modalities in the evaluation. Reviewer **uwkF** highlighted Alchemist's ease of implementation, foreseeing wide adoption. Reviewer **d5ZD** acknowledged Alchemist's flexibility and reusability, making it a versatile tool for diverse tasks. Reviewer **b4or** agreed Alchemist enables cost reduction. We have adopted suggested clarifications, improved our literature review, and conducted new experiments, leading to a much stronger draft.

### New Results Included:
* **Comparison to Other Works [Reviewer n36E]:** We compared Alchemist with other works, including ScriptoriumWS and DataSculpt, as shown in Table 1. Results indicate that **Alchemist uses fewer generated programs while achieving comparable performance.** Additionally, Alchemist addresses data modalities beyond text, unlike these works. We cataloged more detailed differences in the threads below.
* **New Modalities [Reviewer n36E, d5ZD]:** Table 2 includes datasets for ECG heartbeat classification and Census income classification, representing time-series and tabular modalities, respectively. Both illustrate Alchemist's ability to address these settings.
* **Incorporating RoboShot [Reviewer uwkF]:** We integrated RoboShot into Alchemist, using GPT-4o to identify and reject spurious correlations. As shown in Table 3, this integration improved average accuracy and worst-group performance, **enhancing robustness to spurious correlations.**
* **Stability of Generated Programs [Reviewer d5ZD]:** We varied the temperature in GPT-4 API calls, **showing consistent performance** across four datasets, confirming the stability of generated programs (see Table 4).
* **Labeling on Test Set [Reviewer b4or]:** Table 5 presents performance results using the resulting weak supervision label models and trained models (on the annotations from the label models) on eight testing datasets. We generated 10 programs from GPT-3.5-Turbo for each. Results demonstrate that in the majority of scenarios, **using the trained model (with Alchemist annotations) is preferable**.

We have addressed reviewers' questions and placed our comments in their respective threads below. Thank you again for your questions and thoughtful reviews!

---

### Decision · Program_Chairs · 2024-09-25

**Decision:**

Accept (spotlight)

**Comment:**

We thank the authors for their submission and constructive engagement throughout the rebuttal phase of the process. The reviewers and AC were impressed by the simplicity, practicality, and versatility of the contributions, which we believe will be of significant interest to the broader applied ML community. The reviewers also highlight the rigour of the evaluations and clarity of exposition throughout. The authors do a good job addressing key outstanding questions by the reviewers throughout the rebuttal period. We encourage the authors to incorporate these additions in their next revision.